# Mouse aging cell atlas analysis reveals global and cell type-specific aging signatures

Martin Jinye Zhang[1,2,3]*, Angela Oliveira Pisco[4]*, Spyros Darmanis[4], James Zou[1,4,5]*

[1]Department of Electrical Engineering, Stanford University, Palo Alto, United States; [2]Department of Epidemiology, Harvard T.H. Chan School of Public Health, Boston, United States; [3]Program in Medical and Population Genetics, Broad Institute of MIT and Harvard, Cambridge, United States; [4]Chan-Zuckerberg Biohub, San Francisco, United States; [5]Department of Biomedical Data Science, Stanford University, Palo Alto, United States

**Abstract** Aging is associated with complex molecular and cellular processes that are poorly understood. Here we leveraged the Tabula Muris Senis single-cell RNA-seq data set to systematically characterize gene expression changes during aging across diverse cell types in the mouse. We identified aging-dependent genes in 76 tissue-cell types from 23 tissues and characterized both shared and tissue-cell-specific aging behaviors. We found that the aging-related genes shared by multiple tissue-cell types also change their expression congruently in the same direction during aging in most tissue-cell types, suggesting a coordinated global aging behavior at the organismal level. Scoring cells based on these shared aging genes allowed us to contrast the aging status of different tissues and cell types from a transcriptomic perspective. In addition, we identified genes that exhibit age-related expression changes specific to each functional category of tissue-cell types. Altogether, our analyses provide one of the most comprehensive and systematic characterizations of the molecular signatures of aging across diverse tissue-cell types in a mammalian system.

**\*For correspondence:**
jinyezhang@hsph.harvard.edu (MJZ);
angela.pisco@czbiohub.org (AOP);
jamesyzou@gmail.com (JZ)

**Competing interests:** The authors declare that no competing interests exist.

## Introduction

Aging leads to the functional decline of major organs across the organism and is the main risk factor for many diseases, including cancer, cardiovascular disease, and neurodegeneration (*Niccoli and Partridge, 2012*; *López-Otín et al., 2013*). Past studies have highlighted different hallmarks of the aging process, including genomic instability, telomere attrition, epigenetic alterations, loss of proteostasis, deregulated nutrient sensing, mitochondrial dysfunction, cellular senescence, stem cell exhaustion, and altered intercellular communication (*López-Otín et al., 2013*; *Campisi, 2013*; *Vijg and Suh, 2013*; *Nikolich-Žugich, 2018*). However, the primary root of aging remains unclear, and the underlying molecular mechanisms are yet to be fully understood.

To gain a better insight into the mammalian aging process at the organismal level, the Tabula Muris Consortium, which we are members of, created the single-cell transcriptomic data set Tabula Muris Senis (TMS) (*Tabula Muris Consortium, 2020*). TMS is one of the largest expert-curated single-cell RNA sequencing (scRNA-seq) data sets, containing over 300,000 annotated cells from 23 tissues and organs of male and female mice (*Mus musculus*). The cells were collected from mice of diverse ages, making this data a tremendous opportunity to study the genetic basis of aging across different tissues and cell types. The TMS data is organized into scRNA-seq expression of different tissue-cell type combinations (e.g., B cells in spleen) via expert annotation and clustering.

The original TMS paper explored primarily the cell-centric effects of aging, aiming to characterize changes in cell type composition within different tissues. Here we provide a systematic gene-centric study of gene expression changes occurring during aging across different cell types. The cell-centric and gene-centric perspectives are complementary, as the gene expression can change within the same cell type during aging, even if the cell type composition in the tissue does not vary over time.

Our analysis focused on the TMS FACS data (acquired by cell sorting in microtiter well plates followed by Smart-seq2 library preparation *Picelli et al., 2014*) because it has more comprehensive coverage of tissues and cell types (*Supplementary file 1*) and is more sensitive at quantifying gene expression levels as compared to the TMS droplet data. As shown in *Figure 1A*, the FACS data was collected from 16 C57BL/6JN mice (10 males, 6 females) with ages ranging from 3 months (20-year-old human equivalent) to 24 months (70-year-old human equivalent). It contains 120 cell types from 23 tissues, totaling 110,096 cells. We also used the TMS droplet data (derived from microfluidic droplets) for those tissues where the data is available, to further validate our findings on an additional data set generated by a different method.

We investigated the comprehensive expression signatures of aging across tissues and cell types in the mouse. We performed systematic differential gene expression (DGE) analysis to identify aging-related genes in 76 tissue-cell type combinations across 23 tissues (*Figure 1B*, *Supplementary file 1*). Furthermore, we characterized both shared and tissue-cell-specific aging signatures. Our study identified global aging genes (GAGs), namely genes whose expression varies substantially with age in most (>50%) of the tissue-cell types. Interestingly, the changes in expression of these genes are highly concordant across tissue-cell types and exhibit strong bimodality. That is, these genes tend to be either downregulated during aging in most of the tissue-cell types or upregulated across the board. We leveraged this coordinated dynamic to construct an aging score based on the GAGs. We found that the aging score is significantly positively correlated with the chronological age, both in the FACS data and in multiple independent data sets. Moreover, the aging score contrasts the aging status of tissue-cell types with different functionalities and turnover rates, shedding light on the heterogeneous aging process across the 76 tissue-cell types. The score distinguished itself by its single-cell resolution and large data scale, as previous works either studied the biological age at an individual-level (*Harries et al., 2011*; *Holly et al., 2013*; *Peters et al., 2015*; *Fleischer et al., 2018*; *Horvath, 2013*; *Jylhävä et al., 2017*; *Petkovich et al., 2017*) or focused on a small number of organs (*Ori et al., 2015*; *Arrojo E Drigo et al., 2019*). Overall, our analysis highlights the power of scRNA-seq in studying aging and provides a comprehensive catalog of aging-related gene signatures across diverse tissue-cell types.

## Results

### Identification of aging-related genes

We considered 76 tissue-cell types in the TMS FACS data, 26 tissue-cell types in the TMS droplet data, and 17 tissues in an accompanying bulk RNA-Seq mouse aging study (*Schaum et al., 2020*) (referred to as the bulk data) with sufficient sample size. We performed DGE analysis for each tissue-cell type separately, treating all cells from the tissue-cell type as samples. We tested if the expression of each gene is significantly related to aging using a linear model treating age as a numerical variable while controlling for sex. We applied an FDR threshold of 0.01 (the number of comparisons corresponds to the number of genes in the tissue-cell type) and an age coefficient threshold of 0.005 (in the unit of log fold change per month, corresponding to 10% fold change from 3 m to 24 m). For details, please refer to the DGE analysis subsection in Materials and methods.

As shown in *Figure 1B*, the number of significantly age-dependent genes per tissue-cell type ranges from hundreds to thousands. Interestingly, most tissue-cell types have more downregulated aging-related genes than upregulated aging-related genes, suggesting a general decrease in gene expression over aging. This downregulation pattern is unlikely to be confounded by technical factors such as sequencing depth because the 18 m/24 m mice were sequenced at a higher depth (*Figure 1—figure supplement 1C–E*). By doing separate DGE analyses using mice from one sex or mice from a subset of age groups (3 m/18 m), we further found that such a downregulation pattern was mostly driven by 24 m mice and was not specific to one sex (*Figure 1—figure supplement 2*). We also observed a similar pattern in the droplet data (*Figure 1—figure supplement 3*).

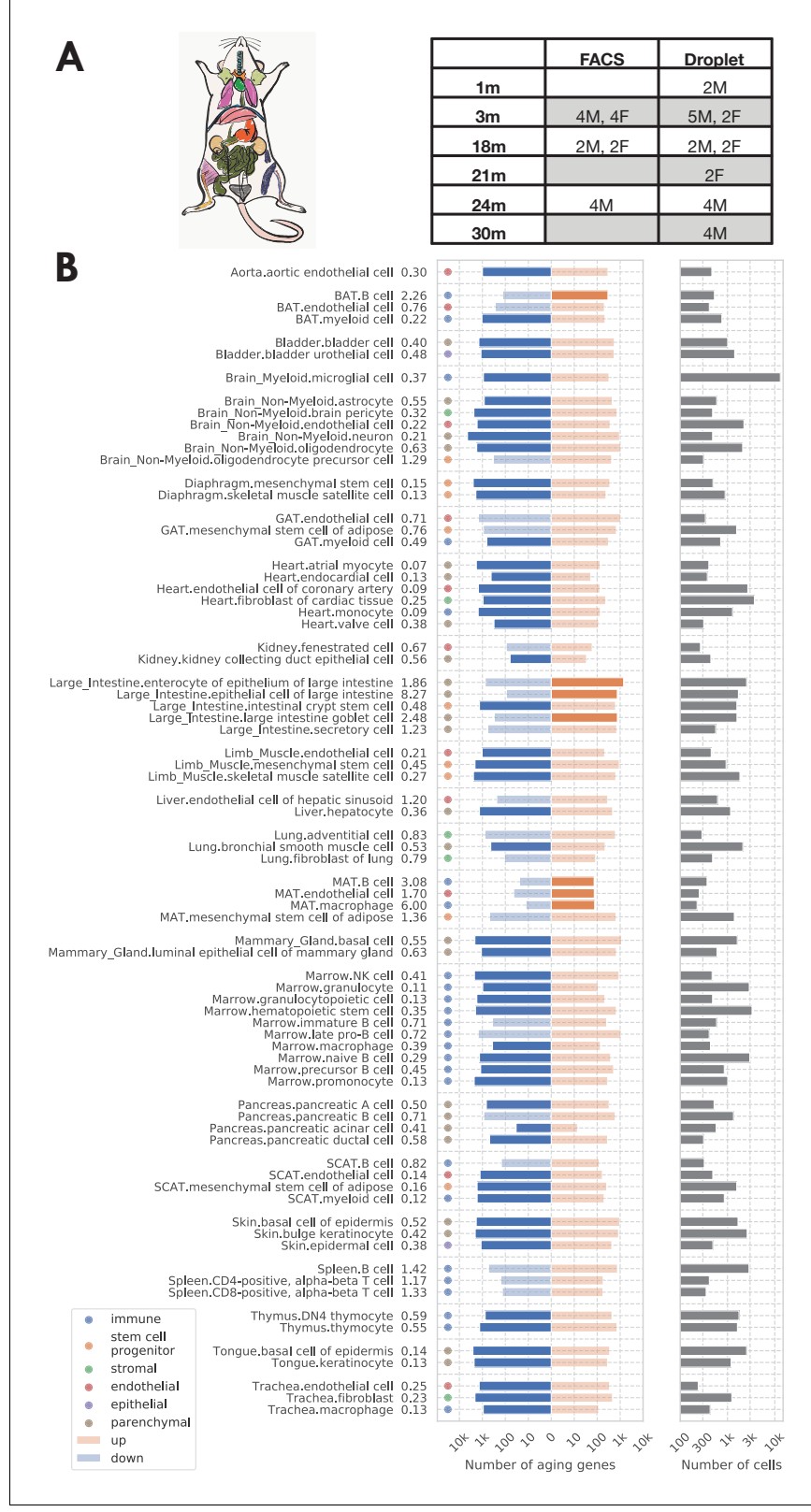

**Figure 1.** Analysis overview. (**A**) Sample description. The TMS FACS data was collected from 16 C57BL/6JN mice (10 males, 6 females) with ages ranging from 3 months (20-year-old human equivalent) to 24 months (70-year-old human equivalent). (**B**) Significantly aging-dependent genes in all 76 tissue-cell types in the FACS data. The left panels show the number of aging-related genes (discoveries) for each tissue-cell type, broken down into the

*Figure 1 continued on next page*

*Figure 1 continued*

number of upregulated genes (orange) and the number of downregulated genes (blue), with the numbers on the left showing the ratio (up/down). Tissue-cell types with significantly more up/downregulated genes (ratio >1.5) are highlighted in solid color. Most tissue-cell types have significantly more downregulated aging genes. The right panel shows the number of cells sequenced for each tissue-cell type.

The online version of this article includes the following source data and figure supplement(s) for figure 1:

**Source data 1.** Source data for *Figure 1*.
**Figure supplement 1.** Covariates for the TMS FACS data and the TMS droplet data.
**Figure supplement 2.** Number of significantly age-dependent genes for each tissue-cell type, from DGE analyses performed using subsets of mice in the TMS FACS data.
**Figure supplement 3.** Number of significantly age-dependent genes for each tissue-cell type, from DGE analyses performed using subsets of mice in the TMS droplet data.
**Figure supplement 4.** Aging trajectory of aging-related genes in the TMS FACS data.
**Figure supplement 5.** Aging trajectory of aging-related genes in the TMS droplet data.
**Figure supplement 6.** Correlations between the age coefficients estimated with and without CDR correction.

In addition, we found that most aging-related genes identified in the analysis have monotonic aging trajectories, meaning that their expressions either increased or decreased monotonically during aging (*Figure 1—figure supplements 4,5*). However, a subset of genes in the FACS data (13%), while being upregulated during aging overall, increased from 3 m to 18 m and slightly decreased from 18 m to 24 m; those genes are enriched in brown adipose tissue (BAT) B cells, large intestine epithelial cells, and mesenchymal adipose tissue (MAT) mesenchymal stem cells of adipose (*Figure 1—figure supplement 4C*).

## Bimodal effects of aging and global aging genes

We found that most genes are significantly related to aging in at least one tissue-cell type (13,376 in the TMS FACS data and 6233 in the TMS droplet data), consistent with the intuition that aging is a highly complex trait involving many biological processes. The aging-related genes discovered in the FACS data significantly overlap with other important gene sets, including both known human and mouse aging markers as recorded in the GenAge database (*Tacutu et al., 2018*), senescence genes (*Campisi, 2013*), transcription factors, eukaryotic initiation factors, and ribosomal protein genes (*Figure 2—figure supplement 1*). Some of the top overlapping genes, significantly related to aging in most tissue-cell types, include known mouse aging markers *Jund*, *Apoe*, and *Gpx4* and known human aging markers *Jund*, *Apoe*, *Fos*, and *Cdc42* from the GenAge database (*Tacutu et al., 2018*), and senescence genes *Jund*, *Junb*, *Ctnnb1*, *App*, and *Mapk1*. In addition, we found that each tissue-cell type has around 5% aging-related genes that are shared by the GenAge human aging markers. However, we did not find any tissue-cell types that are specifically enriched with these known human aging markers, suggesting that the conservation between mouse aging and human aging is relatively uniform across tissue-cell types (*Figure 2—figure supplement 2*).

We visualized all aging-related genes (significant in ≥1 tissue-cell type) in *Figure 2A*, where the color indicates the number of genes. The x-axis shows the weighted proportion of tissue-cell types (out of 76 tissue-cell types) where the gene is significantly related to aging, while the y-axis shows the weighted proportion of tissue-cell types where the gene is upregulated. The tissue-cell type weights used here are inversely proportional to the number of cell types in the tissue, in order to ensure equal representation of the tissues. The visualization makes it clear that there are more downregulated aging genes than upregulated ones, consistent with *Figure 1B*. Perhaps more strikingly, a bimodal pattern is apparent, in the sense that the aging-related genes tend to have a consistent direction of change during aging across most tissue-cell types. Interestingly, it was also recently reported in other studies that many shared aging-related genes exhibit consistent direction of change during aging across mouse tissues and cell types, including the brain (*Ximerakis et al., 2019*), kidney, lung, and spleen (*Kimmel et al., 2019*).

Genes that exhibit age-related expression changes across many cell types have particularly strong bimodal behavior. This motivated us to define *global aging genes* (GAGs), namely the genes that are significantly related to aging in more than 50% of weighted tissue-cell types (i.e., cell types after normalizing by cell type frequencies using the tissue-cell type weights as described above). We

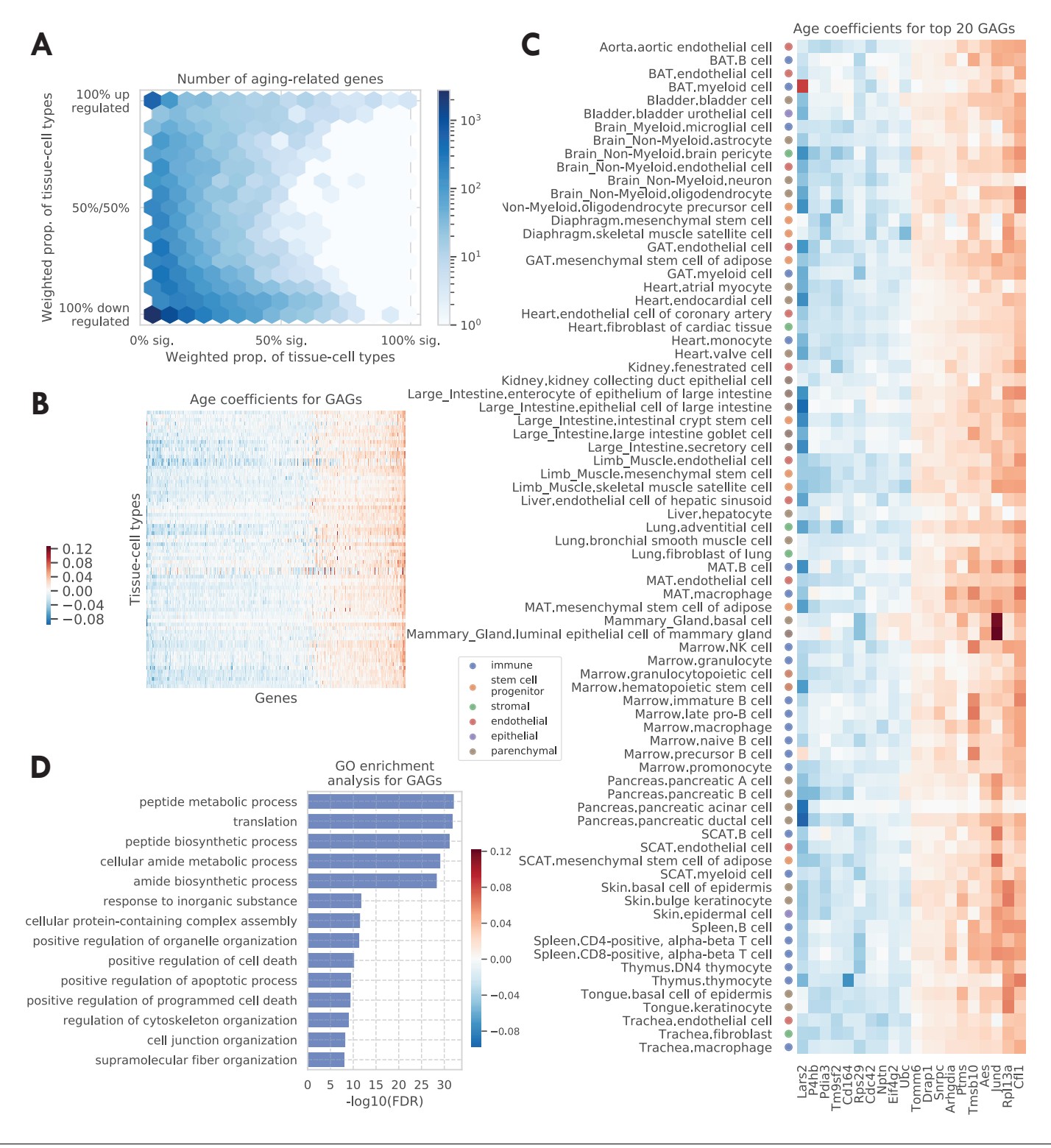

**Figure 2.** Tissue-cell level global aging genes (GAGs). (**A**) Tissue-cell level aging-related genes with color indicating the number of the genes. The x-axis shows the weighted proportion of tissue-cell types (out of all 76 tissue-cell types) where the gene is significantly related to aging, while the y-axis shows the weighted proportion of tissue-cell types where the gene is upregulated. (**B and C**) Heatmap of age coefficients of the GAGs (panel **B**) and the top 20 GAGs (panel **C**). The age coefficients are in the unit of log fold change per month and blue/red represent down/upregulation. (**D**) Top GO biological pathways for the GAGs.

*Figure 2 continued on next page*

eLife Research article

Computational and Systems Biology

*Figure 2 continued*

The online version of this article includes the following source data and figure supplement(s) for figure 2:

**Source data 1.** Source data for *Figure 2*.
**Figure supplement 1.** Comparison between aging-related genes discovered in the TMS FACS data and other gene sets, including the mouse aging genes and human aging genes in the GenAge database (*Tacutu et al., 2018*), senescence genes as provided by the IPA software (*Krämer et al., 2014*), transcription factors, eukaryotic initiation factors, and ribosomal protein genes (Rpl/Rps genes).
**Figure supplement 2.** Overlap between the TMS FACS aging-related genes and the GenAge aging markers.
**Figure supplement 3.** Additional pathway enrichment analysis results.
**Figure supplement 4.** KEGG pathways and IPA pathways for the 330 TMS FACS GAGs.

identified 330 GAGs in total, among which 93 are consistently upregulated and 190 consistently downregulated (>80% of weighted tissue-cell types); only 47 have an inconsistent directionality (upregulated in 20–80% of weighted tissue-cell types). We found that the GAGs significantly overlap with genes known in aging-related diseases, including strong overlap with genes related to Alzheimer's disease (p=2.4e-11), neuroblastoma (p=1.4e-7), fibrosarcoma (p=3.3e-5), and osteoporosis (p=1.5e-4), and relatively weaker overlap with genes related to Huntington's disease (p=2.5e-3), skin carcinoma (p=3.6e-3), kidney cancer (p=1.2e-3), acute promyelocytic leukemia (p=3.2e-3), acute myeloid leukemia (p=3e-3), endometrial cancer (p=1.6e-3), and hypertension (p=1.9e-3) (please see the Global aging genes subsection in Materials and methods for details). Our results are not sensitive to the specific choice of the 50% threshold for selecting GAGs; using different thresholds produced similar gene ontology (GO) enrichment analysis results (*Figure 2—figure supplement 3C*) or GAG scores (*Figure 3—figure supplement 1A–D*) as detailed below.

We visualized the age coefficients for 10 top up/downregulated GAGs (a consistent direction in >80% of weighted tissue-cell types) that are related to aging in the most number of tissue-cell types, as shown in *Figure 2C*. Many of these genes have been previously shown to be highly relevant to aging. For example, the downregulation of *Lars2* has been shown to result in decreased mitochondrial activity and increase the lifespan for *C. elegans* (*Lee et al., 2003*). On the other hand, *Jund* is a proto-oncogene known to protect cells against oxidative stress and its knockout may cause a shortened lifespan in mice (*Laurent et al., 2008*). Moreover, *Rpl13a* was observed to be upregulated in almost all tissue-cell types. As a negative regulator of inflammatory proteins, *Rpl13a* contributes to the resolution phase of the inflammatory response, ensuring that the inflamed tissues are completely restored back to normal tissues. It also contributes to preventing cancerous growth of the injured cells caused by prolonged expression of the inflammatory genes (*Zhou et al., 2015*; *Mazumder et al., 2003*). Therefore, it is interesting to observe the upregulation of *Rpl13a* given that most old mice have severe inflammatory symptoms.

As shown in *Figure 2D*, Gene Ontology (GO) biological pathway enrichment analysis revealed that the 330 GAGs are associated with apoptosis, translation, biosynthesis, metabolism, and cellular organization. These biological processes are highly relevant to aging (*Tower, 2015*; *Anisimova et al., 2018*; *Barzilai et al., 2012*) and are shared across most cell types, consistent with the intuition that GAGs represent the global aging process across tissue-cell types. In addition, the KEGG pathways associated with the GAGs are consistent with the GO terms and additionally highlighted immune-related pathways and multiple aging-related diseases (*Figure 2—figure supplement 4A*). Moreover, the findings were supported by similar analyses on the set of 59 GAGs discovered in the droplet data (*Figure 2—figure supplement 3A B*). We also performed pathway enrichment analysis using the Ingenuity Pathway Analysis (IPA) software (*Krämer et al., 2014*), which confirmed our findings for the biological processes associated with the GAGs (*Figure 2—figure supplement 4B*). Of note is the finding that the mTOR pathway, a known aging-associated pathway, is predicted to be inhibited given the expression of the GAGs (*Weichhart, 2018*; *Papadopoli et al., 2019*; *Johnson et al., 2013*; *Figure 2—figure supplement 4C*). Interestingly, mTOR downregulation has been shown to promote longevity (*Stallone et al., 2019*; *Laplante and Sabatini, 2012*), a further indication that the GAGs are related to the aging process.

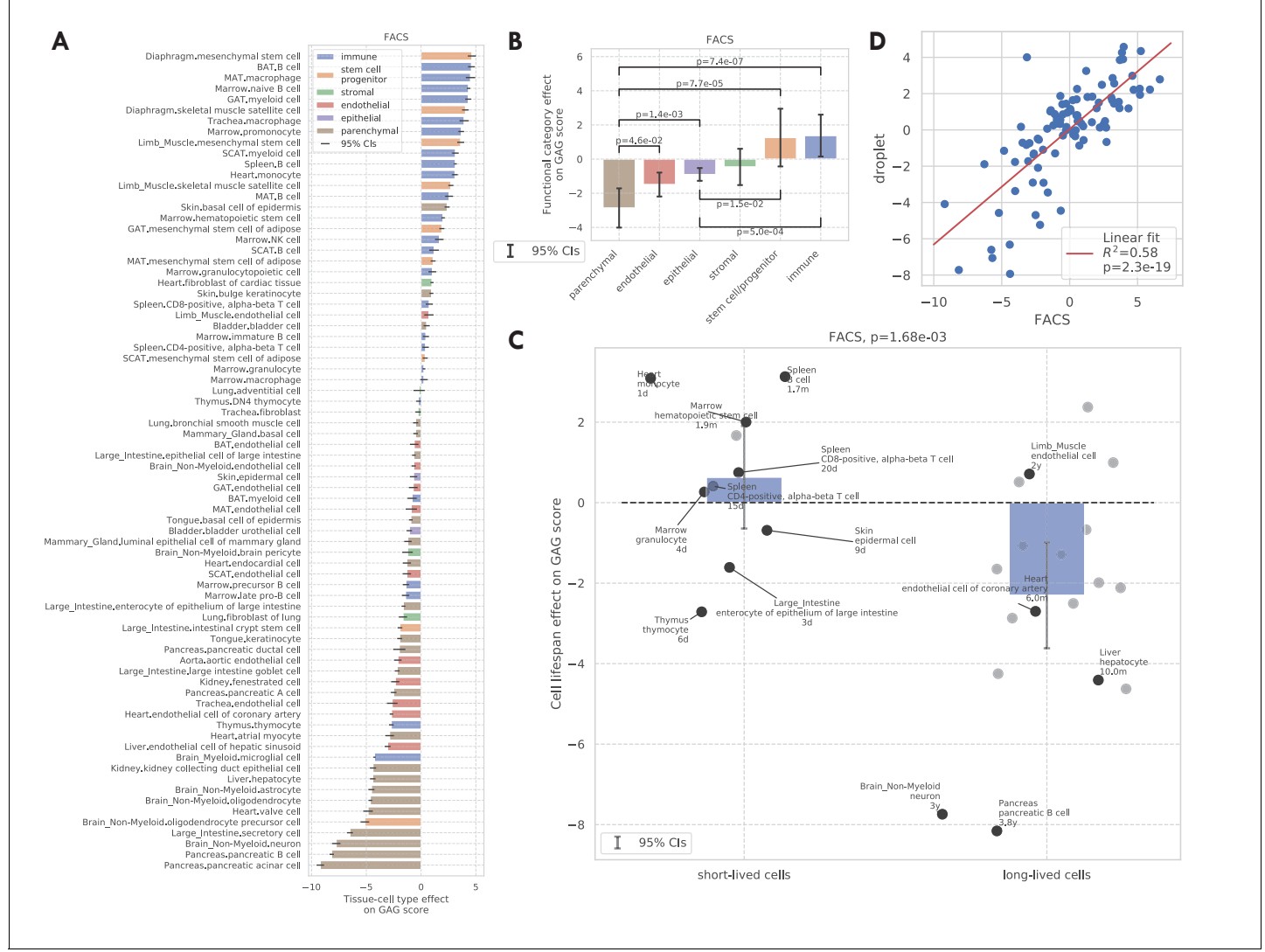

**Figure 3.** GAG score. (**A**) Tissue-cell GAG score effects with 95% confidence intervals. The color represents the functional category of the tissue-cell type. (**B and C**) Effects of cell functional categories (panel **B**) and binary cell lifespan (panel **C**) on the GAG score, meta-analyzed over all tissue-cell types within the group. A positive y-value means that the cells in the group (functional category group for panel **B** and binary cell lifespan group for panel **C**) have higher GAG score values than other cells of the same age and sex. 95% confidence intervals and nominal p-values are provided to quantify the differences between categories. In panel **C**, the average lifespan annotation is also provided for a subset of tissue-cell types where such information is available. (**D**) Comparison between the tissue-cell GAG score effects estimated from the FACS data (x-axis) and the droplet data (y-axis). Each dot corresponds to a tissue-cell type, and a linear fit is provided showing that the estimates are consistent.

The online version of this article includes the following source data and figure supplement(s) for figure 3:

**Source data 1.** Source data for *Figure 3*.
**Figure supplement 1.** Additional validations of the GAG score.
**Figure supplement 2.** Additional validations of the GAG score.
**Figure supplement 3.** Distribution of GAG scores across cells; stratified by tissue-cell types and grouped by mice's chronological age.
**Figure supplement 4.** Tissue-cell GAG score effects for the four validation data sets.
**Figure supplement 5.** Tissue-level analysis as validation.
**Figure supplement 6.** Number of discoveries for each tissue.
**Figure supplement 7.** Tissue-cell validations.
**Figure supplement 8.** Tissue-level GAG score effects.

## GAG score contrasts the heterogeneous aging status of tissue-cell types

Following the analysis of GAGs, we next leveraged these marker genes to characterize the holistic aging status of different tissue-cell types. We aggregated the expression of global aging markers into a single score for each cell, referred to as the GAG score (please see the GAG score subsection in Materials and methods for details). We used the FACS data to identify the GAGs because it has more comprehensive coverage of different tissue-cell types. Intuitively, the GAG score tags the global aging process, and it reflects both the chronological age of the organism and the tissue-cell type-specific aging effects. To formally dissect these components, we defined a fixed-effect model with the GAG score being the response variable and various other factors, including the chronological age, sex, and binary-coded tissue-cell types, being explanatory variables. As a sanity check, the chronological age effect on the GAG score is significantly positive (p<1e-100). The model explains 60.2% of the GAG score variance in the TMS FACS data while adding extra interaction terms between age and each tissue-cell type only slightly increased the model fit (explained variance 62.6%), indicating a reasonable fit for the current model (*Figure 3—figure supplement 2C*). Interestingly, while most young cells have smaller GAG scores and old cells have larger GAG scores, we also found four tissue-cell types whose GAG scores are similar between age groups and eight tissue-cell types that have a subpopulation of cells whose GAG scores are more similar to that of cells from a different age group (*Figure 3—figure supplement 3*), highlighting the heterogeneity of the aging process across tissue-cell types.

We next considered the tissue-cell type effects on the GAG score. Intuitively, a larger GAG score effect suggests that the corresponding cell type could be molecularly more sensitive to aging compared to other cells in the same animal. As shown in *Figure 3A*, immune cells and stem cells have higher GAG score effects, while most parenchymal cell types have lower GAG score effects; such a contrast is also statistically significant, as shown in *Figure 3B*. Indeed, immune cells and stem cells are known to undergo the most substantial changes with aging. Specifically, the aging of the immune system is commonly linked to the impaired capacity of elderly individuals to respond to new infections (*Montecino-Rodriguez et al., 2013*). Also, adult stem cells are critical for tissue maintenance and regeneration, and the increased incidence of aging-related diseases has been associated with a decline in the stem cell function (*Ermolaeva et al., 2018*). On the other hand, parenchymal cells like pancreatic cells, neurons, heart myocytes, and hepatocytes have lower aging scores. This could be an indication that these tissue-specialized cell types are more resilient to aging and are able to maintain their functions despite the changes in the animal.

We also found that the tissue-cell GAG score effects are in general positively correlated with the cell turnover rate. For example, short-lived cells like skin epidermal cells, monocytes, and T cells (*Koster, 2009*; *Guilliams et al., 2018*; *Westera et al., 2013*) have higher GAG score effects while long-lived cells like neurons, oligodendrocytes, pancreatic β-cells, liver hepatocytes, and heart atrial myocytes (*Arrojo E Drigo et al., 2019*; *Teta et al., 2005*; *Magami et al., 2002*; *Bergmann et al., 2015*) have very low GAG score effects. To quantify this observation, we assigned a binary cell lifespan label to a subset of cell types where such data is available from literature (*Milo et al., 2010*; *Hobson and Denekamp, 1984*; *Stewart et al., 1980*; *Lawson et al., 1992*; *Arrojo E Drigo et al., 2019*; *Darwich et al., 2014*; *Lowry and Zehring, 2017*; *Geering et al., 2013*; *Cheshier et al., 1999*; *Fulcher and Basten, 1997*; *Scollay et al., 1980*) (long for >180 days and short for <90 days, *Supplementary file 1*); using binary labels instead of the actual values allows us to incorporate more cell types whose exact lifespan information is not available but are known to be long-/short-lived. We found that short-lived cells have significantly higher GAG score effects than long-lived cells (p=1.68e-3, *Figure 3C*). One possible explanation is that the GAG score is associated with the biological processes related to cell proliferation, development, and death, which are more active in cell types that have a higher turnover rate. This striking difference is also consistent with the intuition that cells that have undergone more divisions (also with higher turnover rates) are ''older' and could have molecular memories.

## Validating the GAG score on external data

We performed several analyses to validate the robustness of the GAG score. First, the GAG score is not sensitive to perturbations of the current scoring method, including using different criterion to

select GAGs or not performing cell-wise background correction (*Figure 3—figure supplement 1A–D*). We also found that estimating tissue-cell GAG score effects using only old cells gave an almost identical result (*Figure 3—figure supplement 1H*).

Next, we performed a parallel analysis to identify GAGs on the TMS droplet data and found 59 such genes (due to smaller sample size and detection power). We found that 34 genes were shared between the droplet GAGs and the 330 FACS GAGs as described above (p=9e-48). Similar to the FACS GAGs, we also found that the droplet GAGs significantly overlap with genes known in many aging-related diseases, including Alzheimer's disease (p=2.8e-4), neuroblastoma (p=1.2e-3), and fibrosarcoma (p=1.4e-3). In addition, we considered a set of 261 shared aging genes reported in *Kimmel et al., 2019*. This scRNA-seq study contains cells from the kidney, lung, and spleen in both young and old mice, and the 261 shared aging genes were defined as genes significantly related to aging in more than five cell types in the paper (*Kimmel et al., 2019*). We found that 90 genes were shared between Kimmel et al. genes and the FACS GAGs (p=2e-105) and 42 genes were shared between Kimmel et al. genes and the droplet GAGs (p=7e-69). All p-values reported here were computed via Fisher's exact tests.

Using the GAGs identified from the TMS FACS data, we computed the GAG score and further estimated the GAG score effects for cells in the TMS droplet data, the bulk data (treating each mouse sample as a ''cell''), the Kimmel et al. data, and the data set from *Kowalczyk et al., 2015*. This last data set has only three subtypes of hematopoietic stem cells (HSCs) and was therefore omitted in other analyses. We found that the chronological age effect on the GAG score is significantly positive in all four validation data sets (p=7e-10 for the bulk data due to smaller sample size and detection power, and p<1e-100 for the other three data sets). Since the GAGs were selected based on the FACS data, the GAG score is agnostic of the age labels in the four validation data sets, confirming that the GAG score is truly indicative of the aging process.

The tissue-cell GAG score effects estimated from the other four data sets are also in line with those estimated from the FACS data (*Figure 3—figure supplements 1,4E F*). Specifically, in the droplet data and the Kimmel et al. data, immune cell types have higher GAG score effects while epithelial, endothelial, and parenchymal cell types have lower GAG score effects. In particular for the droplet data, short-lived cell types also have higher but non-significant GAG score effects, due to a smaller number of annotated cell types. While looking at the bulk data, we found that immune-related tissues and organs such as whole blood, spleen, and marrow have the highest GAG score effects. It is interesting to observe that in the Kowalczyk et al. data, the GAG score effects of MPPs (multipotent progenitors) is less than that of ST-HSCs (short-term HSCs) which is less than that of LT-HSCs (long-term HSCs). This is exactly aligned with the differentiation potentials of these three cell types, consistent with the hypothesis that more stem-like cells have higher GAG score effects as observed in the FACS data.

Finally, we found that the tissue-cell GAG score effects are highly consistent between data sets (correlation 0.76 with p=2e-19 between the FACS data and the droplet data in *Figure 3D*, correlation 0.75 with p=1e-3 between the FACS data and the Kimmel et al. data in *Figure 3—figure supplement 1G*). In summary, we showed that the GAG score is capable of describing the chronological age as well as the transcriptional changes during the aging process. Furthermore, we could use the tissue-cell GAG score effects to contrast the aging status of cell types with different biological properties, including functional categories and turnover rates. This provides a comprehensive analysis demonstrating how the GAG score captures the heterogeneous molecular effects of aging. Additionally, we repeated both the DGE analysis and the GAG score analysis at the tissue-level by combining all cell types from the same tissue. We observed qualitatively similar findings, supporting the robustness of the analyses. Please see *Figure 3—figure supplements 5–8* for more details.

## Category-specific aging genes

We next consider genes specific to a subset of tissue-cell types, including functional-category-specific genes, cell type-specific genes, tissue-specific-genes, and tissue-cell type-specific genes. Given a set of tissue-cell types, in order to have an overall meta age coefficient for cells in this tissue-cell type set, we first combined the age coefficients of all tissue-cel types within the set by meta-analysis; similarly, we also computed the outside-set meta age coefficient by meta-analyzing all outside-set tissue-cell types. Then we selected the genes that have significantly different within-set and outside-set meta age coefficients as the set-specific genes (please see the Category-specific aging

genes subsection in Materials and methods for more details). Of note, almost no genes identified here are shared by GAGs.

In the original TMS paper (*Tabula Muris Consortium, 2020*), each tissue-cell type was assigned one of the six functional category labels, namely endothelial, epithelial, immune, stem/progenitor, stromal, and parenchymal cells. When examining the data by functional category, we found that the endothelial, immune, stem, and stromal cells exhibit highly category-specific aging behavior. Indeed, we found a higher number of specific aging genes for these categories (*Figure 4B*). Moreover, their age coefficients are specific to the respective functional category, as we can see from the clear block structure across tissue-cell types (*Figure 4A*). In addition, when performing GO biological pathway enrichment analysis on these six sets of genes separately, we only found significant pathways for

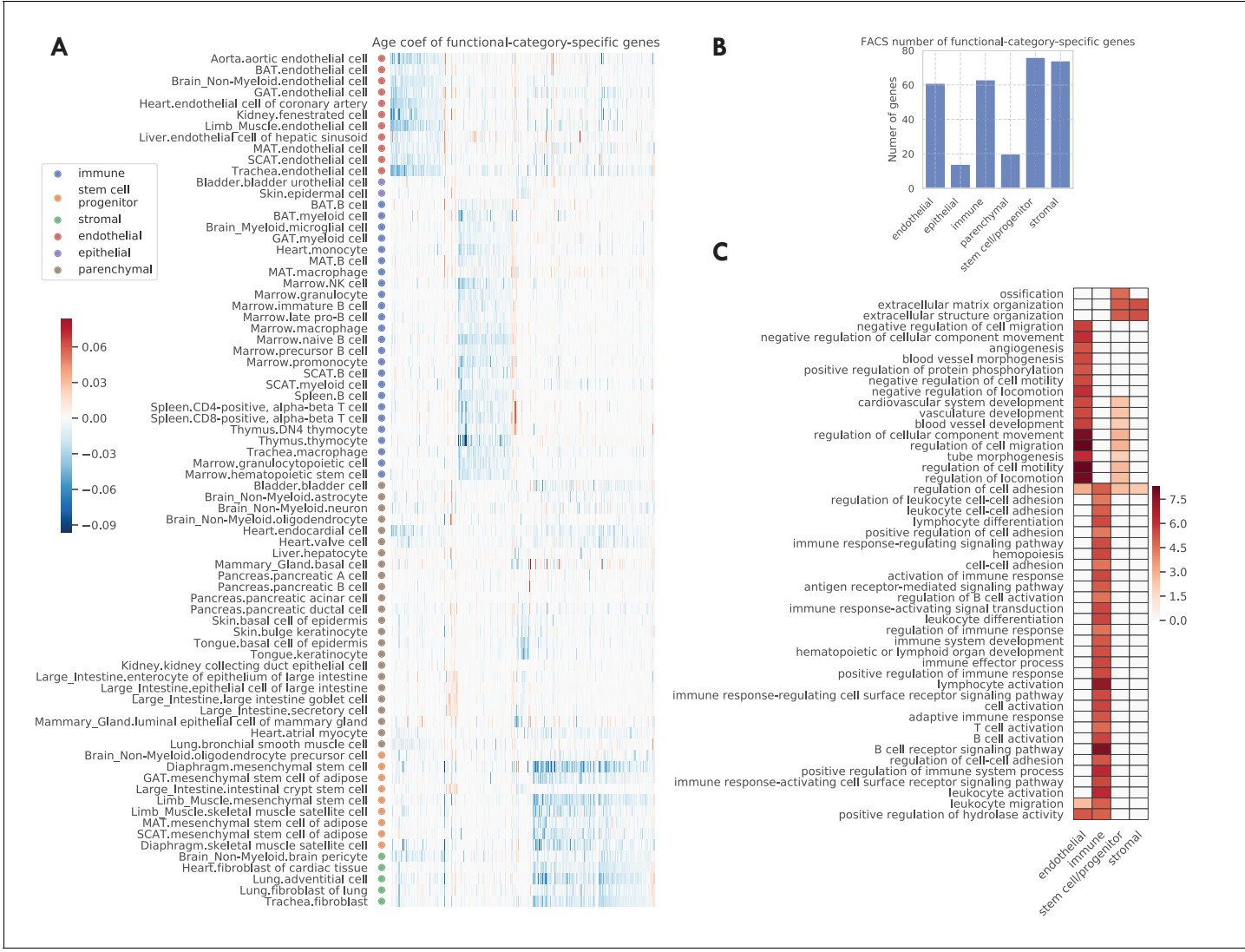

**Figure 4.** Functional-category-specific genes. (**A**) Age coefficients, in the unit of log fold change per month, for functional-category-specific genes. Both genes in the x-axis and tissue-cell types in the y-axis are ordered by functional categories. For example, the upper-left block corresponds to endothelial-specific genes. (**B**) Number of functional-category-specific genes for each category. (**C**) GO biological pathways for functional-category-specific genes, with color representing the negative log10 FDR.

The online version of this article includes the following source data and figure supplement(s) for figure 4:

Source data 1. Source data for *Figure 4*.
Figure supplement 1. Age coefficients of top functional-category-specific genes.
Figure supplement 2. Two examples of functional-category-specific genes.
Figure supplement 3. Cell type-specific genes.

these four categories (*Figure 4C*). Among them, endothelial-specific genes were associated with various processes related to angiogenesis and negative regulation of cell migration; the latter suggests decreased endothelial cell functionality during aging because endothelial cell migration is essential to angiogenesis (*Lamalice et al., 2007*). Also, immune-specific genes were associated with activation of various immune responses, in line with a strong link between the aging process and the immune system (*Tabula Muris Consortium, 2020*; *Nikolich-Žugich, 2018*). In addition, stem-specific genes were associated with ossification and diverse angiogenesis processes, and both stem-specific and stromal-specific genes were associated with extracellular matrix and structure organization.

Such an analysis also facilitates the discovery of interesting specific genes related to aging. For example, *C2cd4b* (*Figure 4—figure supplements 1* and *2A*), a parenchymal-specific gene, has large age coefficients in several pancreatic, mammary gland, large intestine cell types, and almost zero age coefficients in other cell types. The increased expression of *C2cd4b* has been associated with an increased risk of type 2 diabetes (*Kycia et al., 2018*) and increased expression of *C2cd4b* in old mice pancreatic cell types may suggest the increased risk of type 2 diabetes for these mice. In addition, *C2cd4b* has been shown to lead to sexually dimorphic changes in body weight and glucose homeostasis (*Mousavy Gharavy et al., 2021*), in line with the fact that the mammary gland is a well-known sexually dimorphic tissue. A second example is *Gsn* (*Figure 4—figure supplements 1* and *2B*), which is downregulated in stromal and stem cell types and upregulated in other cell types during aging. This can be explained by its function for making gelsolin, an important protein for cell movement. Not only is cell movement important to immune and endothelial cells, but *Gsn* has also been shown to be a potential biomarker to aging-related neurodegeneration (*Manavalan et al., 2013*).

Beyond functional-category-specific genes, we also identified genes specific to several cell types, including B cells, basal cells of epidermis, endothelial cells, macrophages, mesenchymal stem cells, mesenchymal stem cells of adipose, myeloid cells, and skeletal muscle satellite cells, corroborated by their association with related biological processes (*Figure 4—figure supplement 3*). The method also allowed us to identify genes specific to each tissue. However, we did not find any genes specific to a single tissue-cell type. All the gene sets are available in *Supplementary file 3*.

## Discussion

This study provides a systematic and comprehensive analysis of aging-related transcriptomic signatures by analyzing 76 tissue-cell types in the TMS FACS data. Together with the analysis in the first publication of Tabula Muris Senis (*Tabula Muris Consortium, 2020*), it forms one of the largest analysis to date of the mammalian aging process at the single-cell resolution. Of particular interest are the 330 global aging genes (GAGs) identified in the study. These genes exhibit aging-dependent expressions in a majority of tissue-cell types in the mouse. The GAGs are enriched with many interesting genes, including known human and mouse aging markers, aging-related disease genes, senescence genes, transcription factors, eukaryotic initiation factors, and ribosomal protein genes. Interestingly, most of the GAGs are strongly bimodal as their expressions either decrease or increase during aging across almost all tissue-cell types, suggesting that these genes have a uniform response to aging, which is robust to the specific tissue or cellular context. Moreover, we find a systematic decrease in expression for most genes as well as a decrease in the number of actively expressed genes, suggesting a turning off of transcription activity as the animal ages. A recent study has observed that the number of expressed genes decreases during cellular differentiation in mouse (*Gulati et al., 2020*). It is interesting that we quantify a similar phenomenon for aging, despite the substantial longer time-scale of aging compared to differentiation.

While we focused on detecting genes whose changes with age are linear and non-sex-specific, it is also interesting to study aging-related changes that are non-linear or sex-specific. For example, as shown in *Figure 1—figure supplement 1*, the number of expressed genes changes with age in a non-linear and sex-specific manner, suggesting the existence of such genes. We have validated our findings using the TMS droplet data, the bulk RNA-seq data, and external data sets, and it is important to have further validations in future studies. In particular, the bimodal expression pattern is less apparent in the TMS droplet data, perhaps due to its limited tissue-cell type coverage and relatively shallower sequencing depth.

The remarkable bimodal consistency of the GAGs makes them useful as biomarkers to characterize the aging status of individual cells. We proposed a new aging score, namely the GAG score, based on the GAGs. The tissue-cell type-specific GAG score effects quantify how sensitive each tissue-cell type is to aging and are positively correlated with the cell division rate. For example, immune cells tend to have higher GAG scores than other cells of the same age and sex, which reflects the phenomenon that they undergo many cycles of cell division and also change substantially during the animal's lifespan. One hypothesis is that the GAG score captures some aspects of the true biological age of the cells, which could be different from the birth age of the animal. An interesting direction of future work is to further investigate this model with functional experiments. In line with this, it would be important to study how some of the transcriptomic changes we quantify here, for example the downregulation of mTOR, point toward healthy aging or how can they inform experiments that can uncover the mechanism to ameliorate the aging effects. In addition, while the GAG score was proposed to capture the global aging status, there also exist cell type-specific aging programs; a combination of both would better characterize the overall aging status of a cell. The construction of cell type-specific aging scores would require a more comprehensive longitudinal catalog of diverse subtypes and states of cells within different cell types. While this is beyond the scope of the current work, it is an exciting direction to pursue when such data becomes available.

The GAG score is also related to the transcriptome age predictors developed in previous works (*Harries et al., 2011*; *Holly et al., 2013*; *Peters et al., 2015*; *Fleischer et al., 2018*), in the sense that they all use the gene expression information and are predictive of the animals' chronological age. The commonly used approach in previous works is to train a model (e.g., linear/logistic regression model) to predict the individuals' chronological age from their gene expression. Instead of a model-fitting algorithm, our GAG score uses the GAGs that were selected from a broad range of tissue-cell types in an unbiased manner, by meta-analyzing the DGE results of 76 tissue-cell types and putting each tissue-cell type on the same footing. This ensures that the genes used by the GAG score capture the shared aging process and are not biased toward certain tissue-cell types. Indeed, the GAGs were shown to be associated with biological processes that are highly relevant to aging (*Figure 2*), providing better interpretability of the score. In comparison, previous studies focused on only one specific tissue, such as the blood (*Harries et al., 2011*; *Holly et al., 2013*; *Peters et al., 2015*) or dermal fibroblasts (*Fleischer et al., 2018*), and hence may have selected genes that were biased toward that particular tissue.

Overall, our study provides a comprehensive characterization of aging genes across a wide range of tissue-cell types in mice. In addition to the biological insights, it also serves as a comprehensive reference for researchers working on related topics.

## Materials and methods

### Data preprocessing

We considered five data sets, namely the TMS FACS data, the TMS droplet data, the data in *Schaum et al., 2020* (referred to as the bulk data), the data in *Kimmel et al., 2019*, and the data in *Kowalczyk et al., 2015*. For the TMS FACS data and the TMS droplet data, we filtered out genes expressed in fewer than 3 cells, filtered out cells expressing fewer than 250 genes, and discarded cells with a total number of counts fewer than 5000 for the FACS data and a total number of unique molecular identifiers (UMIs) fewer than 2500 for the droplet data. For the bulk data, we filtered out genes expressed in fewer than five samples, and filtered out samples expressing fewer than 500 genes. We did not filter cells for the other two data sets. For all five data sets, we normalized each sample to have 10,000 reads/UMIs per sample, followed by a log transformation ($\log(x + 1)$ where $x$ is the read count). We note that such a procedure is the same as that in the original paper (*Tabula Muris Consortium, 2020*). We did not correct for batch effects as no substantial batch effects were identified in the original TMS paper (*Tabula Muris Consortium, 2020*).

### DGE analysis

As shown in *Supplementary file 1*, we considered 76 tissue-cell types in 23 tissues with more than 100 cells in both young (3 m) and old (18 m, 24 m) age groups for the TMS FACS data; 26 tissue-cell types in 11 tissues with more than 500 cells in both young (1 m, 3 m) and old (18 m, 21 m, 24 m, 30

m) age groups for the TMS droplet data; and all 17 tissues for the bulk data (*Schaum et al., 2020*). We required more cells for the TMS droplet data than the TMS FACS data because the droplet data has a much lower sequencing depth (6000 UMIs per cell, as compared to 0.85 million reads per cell for the FACS data). Also, we did not focus on the TMS droplet data in the main results due to its limited tissue and cell type coverage.

We performed a DGE analysis for cells in each tissue-cell type separately. In the DGE analysis for a tissue-cell type, all cells in the tissue-cell type were treated as samples, and a separate test was performed for each gene with the observations being the expressions of the gene across the cells. We identified genes significantly related to aging using a linear model treating age as a continuous variable while controlling for sex, namely,

$$\text{gene expression} \sim \text{age} + \text{sex}. \tag{1}$$

Since the tests were performed at a cell level for each tissue-cell type, the cell numbers will only affect the detection power but will not bias the result. Therefore, it is not a confounding factor and was not controlled for. We used the MAST package (*Finak et al., 2015*) (version 1.12.0) in R to perform the DGE analysis. The zero counts in the scRNA-seq data were handled by the MAST package and we do not observe other types of missing data. We did not control for the cellular detection rate (*Finak et al., 2015*) (CDR, corresponding to the number of expressed genes in a cell) because we found that in our data, CDR is positively correlated with age and negatively correlated with technical covariates such as sequencing depth and the number of detected ERCC spike-ins, both when considering all the cells or when stratified by sex (*Figure 1—figure supplement 1*). As a result, controlling for CDR may remove genuine aging effects. We note that CDR is defined as the number of expressed genes in a cell, which is a fundamental quantity of the data set and is not specific to the MAST package. Therefore, such an observation does not imply a potential issue of MAST. Nonetheless, we found that the age coefficients, estimated with and without CDR correction, were highly correlated (0.89 for the FACS data and 0.93 for the droplet data, *Figure 1—figure supplement 6*), ruling out the possibility that CDR correction would significantly alter the result.

We used the Benjamini–Hochberg (FDR) procedure (*Benjamini and Hochberg, 1995*) to control for multiple comparisons, where the number of comparisons corresponds to the number of genes in the tissue-cell type. We applied an FDR threshold of 0.01 and an age coefficient threshold of 0.005 (in the unit of log fold change per month, corresponding to around 10% fold change from 3 m to 24 m) for detecting genes significantly related to aging.

## Global aging genes

We selected a gene as a GAG if it is significantly related to aging in more than 50% of weighted tissue-cell types. Here, the tissue-cell type weights are inversely proportional to the number of cell types in the tissue, in order to ensure equal representation of tissues.

For the overlap between GAGs and the genes known in aging-related diseases, we considered the top 25 aging-related diseases and obtained their related genes from the Human Disease Database (MalaCards) (*Rappaport et al., 2013*; *Rappaport et al., 2014*; *Rappaport et al., 2017*). We then converted the human genes to the corresponding mouse orthologs using g:Profiler (*Raudvere et al., 2019*) (version 1.2.2). The p-values quantifying the significance of the overlap were computed using Fisher's exact tests.

## Pathway enrichment analysis

We used g:Profiler (*Raudvere et al., 2019*) to perform GO biological pathway enrichment analysis. We considered biological pathways with FDR smaller than 0.01. We used Gene Set Enrichment Analysis (GSEA MGSig Database) (*Subramanian et al., 2005*; *Liberzon et al., 2011*) to perform the KEGG pathway analysis. We filtered for mouse genes and considered biological pathways with FDR smaller than 0.05. We also used the IPA software (*Krämer et al., 2014*) to perform canonical pathway analysis (*Figure 2—figure supplement 4*). For *Figure 2—figure supplement 4A*, we used an FDR threshold of 1e-5 and a z-score threshold of 0.5.

## GAG score

Given a set of GAGs (e.g., the FACS GAGs), the cell-wise GAG score for cell $i$ is computed as:

1. Compute the raw GAG score as the average expression of the upregulated GAGs (significantly related to aging in >50% of weighted tissue-cell types and upregulated in >80% of weighted tissue-cell types) minus the average expression of the downregulated GAGs (significantly related to aging in >50% of weighted tissue-cell types and downregulated in >80% of weighted tissue-cell types), That is,

$$\mathrm{Raw\,GAG\,score}_i = \mathrm{mean(upregulated\,GAGs)}_i - \mathrm{mean(downregulated\,GAGs)}_i \tag{2}$$

2. Following the recipe of *DeTomaso et al., 2019*, z-normalize the raw GAG score using the expected mean and variance of a random set of genes with the same number of up/downregulated genes:

$$\mathrm{GAG\,score}_i = \mathrm{Raw\,GAG\,score}_i / \left[ \mathrm{std}_i * \sqrt{1/n_{\mathrm{up}} + 1/n_{\mathrm{down}}} \right], \tag{3}$$

where $\mathrm{std}_i$ is the standard deviation of the gene expression of cell $i$, $n_{\mathrm{up}}$ is the number of upregulated GAGs, and $n_{\mathrm{down}}$ is the number of downregulated GAGs.

For robustness consideration, we checked the expression levels of the genes used for computing the GAG score and found that there are no extreme values. Therefore, the GAG score is unlikely to be dominated by a few highly expressed genes. We also considered another version of GAG score where each gene is weighted by its expression range. We found the two versions produce highly correlated results. Please see *Figure 3—figure supplement 2A B* for more details.

For estimating the GAG score effects, we model the GAG score of a cell $i$ as being linearly dependent of the chronological age, sex, and tissue-cell type of the cell, namely,

$$\mathrm{GAG\,score}_i \sim \mathrm{age}_i + \mathrm{sex}_i + \sum_j \mathrm{tissue-celltype}_{ij} \tag{4}$$

Here, $\mathrm{age}_i$ is the age of the animal (in months) that cell $i$ comes from, $\mathrm{sex}_i$ is one if cell $i$ comes from a male and zero otherwise, and $\mathrm{tissue-celltype}_{ij}$ is one if cell $i$ belongs to tissue-cell type $j$ and zero otherwise. We do not include the intercept term because all binary coded tissue-cell types sum up to one. Finally, we further center both response and explanatory variables and perform an ordinary least square regression to estimate the GAG score effects for the age, sex, and each tissue-cell type.

Meta-analyses in *Figure 3B and C* and *Figure 3—figure supplement 1E F* were performed assuming a random effect model (*Riley et al., 2011*). For comparisons of the tissue-cell GAG score effects between data sets, namely those in *Figure 3D* and *Figure 3—figure supplement 1G*, we used all tissue-cell types instead of restricting to the 76 TMS FACS tissue-cell types and 26 TMS droplet tissue-cell types. This increased the number of overlapping tissue-cell types between data sets.

## Category-specific aging genes

We considered identifying functional-category-specific genes, cell type-specific genes, tissue-specific genes, and tissue-cell type-specific genes. For a set of tissue-cell types, for example the set of all immune tissue-cell types, the genes specific to the set (or set-specific genes) are selected as follows. For each gene, we first estimate its within-set meta age coefficient by meta-analyzing the age coefficients of all tissue-cell types in the set assuming a random effect model (*Riley et al., 2011*). Specifically, it is done by assuming that there is a meta age coefficient for the set of tissue-types, and the age coefficient for each tissue-cell type in the set is a random variable whose mean is equal to the meta age coefficient of the set. Similarly, we estimate the outside-set meta age coefficient by meta-analyzing all tissue-cell types outside the set. Then, we define the set-specific genes to be genes whose:

1. within-set meta age coefficient is significantly different from its outside-set meta age coefficient (FDR < 0.01);
2. within-set meta age coefficient is large enough (absolute value > 0.005);
3. outside-set meta age coefficient is not significantly different from 0 (FDR > 0.01).

The p-values are computed based on the mean and the standard error assuming a normal distribution, and FDR is computed with respect to all genes.

## Code availability

The code for reproducing all results is at https://github.com/czbiohub/tabula-muris-senis/tree/master/2_aging_signature (*Zhang and Pisco, 2021*; copy archived at swh:1:rev:0fd2ee501f4f3bd0e691b6071aee2c9286f1cf92).

## Acknowledgements

We would like to thank S Quake, R Sinha, R Sit, J Cool, B van de Geijn, H Shi, and X Xu for feedback. MJZ and JZ are supported by NSF CCF 1763191, NIH R21 MD012867-01, NIH P30AG059307, and grants from the Silicon Valley Foundation and the Chan-Zuckerberg Initiative. MJZ is also supported by NIH R01 MH115676.

## Additional information

### Funding

| Funder | Grant reference number | Author |
| --- | --- | --- |
| Chan-Zuckberg Biohu | | James Zou |
| National Science Foundation | CCF 1763191 | James Zou |
| National Institutes of Health | R21 MD012867-01 | James Zou |
| National Institutes of Health | P30AG059307 | James Zou |
| Silicon Valley Community Foundation | | James Zou |
| National Institutes of Health | R01 MH115676 | Martin Jinye Zhang |

The funders had no role in study design, data collection and interpretation, or the decision to submit the work for publication.

### Author contributions

Martin Jinye Zhang, Conceptualization, Formal analysis, Investigation, Visualization, Methodology, Writing - original draft, Writing - review and editing; Angela Oliveira Pisco, Conceptualization, Data curation, Supervision, Investigation, Writing - original draft, Writing - review and editing; Spyros Darmanis, Data curation, Writing - review and editing; James Zou, Supervision, Funding acquisition, Investigation, Methodology, Writing - original draft, Writing - review and editing

### Author ORCIDs

Martin Jinye Zhang https://orcid.org/0000-0003-0006-2466
Angela Oliveira Pisco https://orcid.org/0000-0003-0142-2355
James Zou https://orcid.org/0000-0001-8880-4764

### Decision letter and Author response

Decision letter https://doi.org/10.7554/eLife.62293.sa1
Author response https://doi.org/10.7554/eLife.62293.sa2

## Additional files

### Supplementary files

• Supplementary file 1. Summary of tissues and cell types for TMS FACS data, TMS droplet data, and the bulk data.

- Supplementary file 2. Significantly aging-related genes in each tissue-cell type for TMS FACS data and TMS droplet data.
- Supplementary file 3. Gene sets identified in the study, including GAGs and category-specific genes, for TMS FACS data and TMS droplet data.
- Transparent reporting form

### Data availability

All data can be downloaded at https://figshare.com/articles/dataset/tms_gene_data_rv1/12827615.

The following previously published dataset was used:

| Author(s) | Year | Dataset title | Dataset URL | Database and Identifier |
|---|---|---|---|---|
| The Tabula Muris Consortium | 2020 | Tabula Muris | https://doi.org/10.6084/m9.figshare.12827615.v1 | figshare, 10.6084/m9.figshare.12827615.v1 |

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
