## [Decision Letter]

**Acceptance summary:**

This work presents a highly refined and validated global aging (GAG) score, as well as establishing category-specific murine aging genes, which together provide a comprehensive angle to understand aging at different scales. The GAG score enables capture of tissue-cell-type specific effects, setting more focus on the tissue- and cell type-specific aging status, and enables assessment of the general aging status.

**Decision letter after peer review:**

[Editors’ note: the authors submitted for reconsideration following the decision after peer review. What follows is the decision letter after the first round of review.]

Thank you for submitting your work entitled "Mouse Aging Cell Atlas Analysis Reveals Global and Cell Type Specific Aging Signatures" for consideration by *eLife*. Your article has been reviewed by two peer reviewers, and the evaluation has been overseen by a Reviewing Editor and a Senior Editor. The following individual involved in review of your submission has agreed to reveal their identity: Lei Hou (Reviewer #1).

Our decision has been reached after consultation between the reviewers. Based on these discussions and the individual reviews below, we regret to inform you that your work will not be considered further for publication in *eLife*.

Reviewer #1:

The proposed studies aims to in-depth analyses of aging transcriptomic signatures in mouse both at tissue-cell type and global level from single-cell data. It is an important perspective to understand aging not only for specific tissue-cell type but also the differences among them. One of the main contributions is a novel aging score metric, based on which aging process is compared across tissue-cell type pairs. However, I have several concerns about this aging score and would prefer a revision from the authors.

1. I am confused about the method you used to calculate the aging score.

In main text, the aging score is calculated for each cell, then averaged at tissue-cell type level, and regress out age and sex; while in method for each cell, after adjusting the background, age and sex are regressed out from the raw aging score, then the scores are averaged at tissue-cell type level.

2. For either case, the name of the aging score is misleading, as the residuals after regressing out age and sex, it is not a predictor of aging. Instead, it represents how a specific tissue-cell type is off from the average aging effect across all conditions.

3. If it is actually what authors would love to show, the random effect in each tissue-cell type should be taken into consideration: for cells from different tissue-cell type, their intercepts of the regression model exp~ age+sex+ age*sex could be different, and thus a linear mixed model should be more appropriate in this case. It may explain why in supplementary Figure 9, aging scores from immune cells compared to other cell types of this study show even weaker correlation with that based on blood samples from Peters' work (pvalue 5.4e-4, 0.08, compared to 1.8 e-5).

4. For comparing aging effects purpose, I don't understand why aging scores of cells from the young time point should be included in the final aging score for each tissue-cell type. Firstly, it doesn't represent the aging effect if you also include scores from young cells; Secondly, the scores for each tissue-cell type may be confounded by different percentages of young cells.

Reviewer #2:

In the manuscript titled "Mouse Aging Cell Atlas Analysis Reveals Global and Cell Type Specific Aging Signatures", Zhang and et al. systematically explored the aging-related genes in 76 tissue-cell types from 23 tissues in 10 male and 6 female mice from the Tabula Muris Senis single-cell transcriptomic dataset. The authors used a linear regression model to identify a set of up-/down-regulated aging-related genes, found a general down-regulation of gene expression in most tissue-cell types and revealed sets of aging regulated genes that shared by different tissue-cell types or that specifically enriched in some tissue-cell types. The authors further leveraged the average expression difference in the up- and down-regulated global aging genes, and proposed an aging score defined as a correlated factor that determines the change in different tissue-cell types without effects of age and sex. The manuscript is well organized, the results presented are impressing and the conclusions are pretty attractive. However, the data analyses need substantial revisions to draw any convincing conclusions, specifically, I have a list of concerns as below:

1. General concerns:

a. The sample size may be insufficient (10 males, 6 females) to build linear regression model to determine reliable correlations between certain gene expression with the two factors of age and sex in single-cell level.

b. Since the TMS datasets are curated from a collaboration program with different labs, confounding batch effects should be explicitly evaluated and resolved before any further analysis.

c. Though many platforms and technologies may have built-in strategies to deal with uncertainty of RNA copy number, normalization of single cell transcriptome are still highly recommended in the data processing to control technical inconsistency and calculation of DGE. The MAST package is aware of the problem and use the CDR calculation as alternatives of normalization, which, however, was omitted by the authors when using the package.

d. Beside the validation of the aging score, parallel analyses of droplets datasets should be performed in the same tissue-cell types to validate their findings.

2. Identification of aging-related genes:

a. In the linear model of DGE with age and sex, it is unclear what the observations were in the model, whether the statistical test performed in individual cells across different tissue-cell types; how to deal with missing data; and how to control the effect of cell numbers if the tests are performed in tissue-cell types level.

b. FDR calculation is highly depend on simultaneous consideration of comparisons in multiple tests. It is also unclear what the exact number of comparisons considered in each FDR test.

c. Figure 1B shows number comparison between up- and down-regulated genes related to age regardless of sex, however the authors should clarify whether the sex has effect on the result and a new figure of the comparison considering sex is highly recommended.

3. Tissue-cell level global aging markers:

a. The definition of global aging genes is based on an arbitrary cutoff of half of tissue-cell types without considering the data structure, tissue-cell similarities or proportions of cells determined in mice, which may be conceptually confusing if change happened as data accumulating and new tissue-cell types identified.

b. For those up- or down-regulated genes in specific tissue-cell types, a pickup top 20 genes is not convincing and it is necessary to evaluate the significance of the involvement of those genes in shared biological progresses or functions.

4. Aging score based on global aging genes:

a. The aging score is defined based simply on the potential determinants in the average expression changes between up- and down-regulated global aging genes without the effects from age and sex, which is not intuitively straightforward and confusing, given the ambiguous involvement in biological functions or regulations of the list of genes.

b. The authors claim that aging score is correlated with the cell turnover but failed to show any solid evidence.

5. Tissue-cell-specific aging genes:

a. The authors grouped tissue-cell types into 6 categories based on annotated functionality, and then identified over-represented genes in each category as tissue-cell-specific aging genes. I am wondering why not define the tissue-cell-specific aging genes in original tissue-cell types; the arbitrary grouping may be inaccurate and inevitably reduced the resolution of the tissue-cell specificity.

b. Again, the author failed to establish solid connection between these tissue-cell-specific aging genes and the tissue-cell specific functions or aging progress, which makes the result less reliable.

[Editors’ note: further revisions were suggested prior to acceptance, as described below.]

Thank you for submitting your article "Mouse Aging Cell Atlas Analysis Reveals Global and Cell Type Specific Aging Signatures" for consideration by *eLife*. Your article has been reviewed by 2 peer reviewers, and the evaluation has been overseen by a Reviewing Editor and Jessica Tyler as the Senior Editor. The following individual involved in review of your submission has agreed to reveal their identity: Lei Hou (Reviewer #1).

The reviewers have discussed the reviews with one another and the Reviewing Editor has drafted this decision to help you prepare a revised submission.

Summary:

You have made great efforts to refine and validate their global aging (GAG) score, as well as establish category-specific aging genes, which together provide a comprehensive angle to understand aging at different scales. Still, we have some concerns as below.

Essential changes:

1. GAG score aims to capture common aging states across all cell types. Though authors tried to validate it from different datasets, it would be more convincible if they could show how well GAG score is fitted for GAG score ~ chronological age + sex + tissue-cell-type with both scatter plot and proportion of explained. Alternatively, will it fit better for GAG score ~ chronological age + sex + tissue-cell-type + tissue-cell-type * chronological age, where tissue-cell-type could also affect the slope of age.

2. GAG score, due to its nature of capturing global pattern, may lose the power to identify the aging state of those cell types with the specific program associated with aging. It may be the reason for observations mentioned in line 135-138 and 146-148, specifically in Sup.Figure 11, for Brain.Non-Myeloid. neuron, liver.hepatocyte, mamary_glad.bascal cells, cells of 24 months even have a smaller GAG score than cells of 18 month. However, This may be potentially explained by catogery-specific aging genes later identified. Would tissue-cell type with more category-specific aging genes be more likely to show this heterogenous aging state pattern? Further, an accurate aging score for a specific cell type may be better consisting of two parts, global aging score, and specific aging score.

3. Last session of tissue-level section is not interesting to me, since differential genes at cell type are already accurately defined, and there's no reason to go back to bulk differential signal, which may be confounded by other factors such as cell-type proportions.

---

## [Author Response]

[Editors’ note: the authors resubmitted a revised version of the paper for consideration. What follows is the authors’ response to the first round of review.]

Reviewer #1:The proposed studies aims to in-depth analyses of aging transcriptomic signatures in mouse both at tissue-cell type and global level from single-cell data. It is an important perspective to understand aging not only for specific tissue-cell type but also the differences among them. One of the main contributions is a novel aging score metric, based on which aging process is compared across tissue-cell type pairs. However, I have several concerns about this aging score and would prefer a revision from the authors.1. I am confused about the method you used to calculate the aging score.In main text, the aging score is calculated for each cell, then averaged at tissue-cell type level, and regress out age and sex; while in method for each cell, after adjusting the background, age and sex are regressed out from the raw aging score, then the scores are averaged at tissue-cell type level.2. For either case, the name of the aging score is misleading, as the residuals after regressing out age and sex, it is not a predictor of aging. Instead, it represents how a specific tissue-cell type is off from the average aging effect across all conditions.3. If it is actually what authors would love to show, the random effect in each tissue-cell type should be taken into consideration: for cells from different tissue-cell type, their intercepts of the regression model exp~ age+sex+ age*sex could be different, and thus a linear mixed model should be more appropriate in this case. It may explain why in supplementary Figure 9, aging scores from immune cells compared to other cell types of this study show even weaker correlation with that based on blood samples from Peters' work (pvalue 5.4e-4, 0.08, compared to 1.8 e-5).

Thank you for these suggestions. We agree with the reviewer and have substantially revised our aging score analysis. In the revised workflow, we first compute the aging score for each cell the same way as before, namely by taking the mean expression of the up-regulated global aging genes minus the mean expression of the down-regulated global aging genes. We refer to this quantity as the global aging gene (GAG) score as suggested by comment (2).

Next, as suggested by comment (3), instead of using the aging score residuals to study tissue-cell-specific aging effects, we formally define a fixed effect model to jointly model the effects of the chronological age, sex, and tissue-cell types on the GAG score:

GAG score ~ chronological age + sex + tissue-cell-type

Then we estimate the tissue-cell GAG score effects via regression (please see the GAG score subsection in Methods for more details). The tissue-cell GAG score effects capture how the GAG score of cells from the tissue-cell type deviates from the average GAG score of cells in the same age and sex group. Intuitively, a larger GAG score effect suggests that the corresponding cell type could be molecularly more sensitive to aging compared to other cells in the same animal.

We replicated all of our original findings using this estimated tissue-cell type GAG effects. First, we find that different tissue-cell types have heterogeneous effects on the GAG score, following the same ordering as in our original submission (Original Figure 3A, Revision Figure 3A). Second, stem cells and immune cells have higher effects while parenchymal cells have lower effects (Original Supp. Figure 8, Revision Figure 3B). Third, short-lived cells have higher GAG score effects (Original Figure 3B, Revision Figure 3C); and the estimates are consistent between different datasets (Original Figure 3C, Revision Figure 3D). These results demonstrate that we can reproduce and support our findings with this more principled statistical model of aging scores.

We removed the comparison to Peters et al. work since we do not focus on predicting the chronological age using the GAG score as was done in Peters et al. Instead, we focus on using the GAG score to contrast the tissue-cell specific aging effects. Moreover, Peters et al. primarily focused on blood cell types in their analyses, making it difficult to directly compare with our work which leverages a diverse range of tissue-cell types.

Instead, we included two additional data sets for comparison: Kimmel et al. GR 2019 on mouse aging in lung, kidney, and spleen, and Kowalczyk et al. GR 2015 on mouse aging in subtypes of HSCs. We further validated our GAG score analysis by showing:

1. The global aging genes identified in the Tabula Muris Senis (TMS) FACS data and the TMS droplet data significantly overlap with each other and both sets of genes significantly overlap with the 261 shared aging genes in Kimmel et al. (p=9e-48 between FACS and droplet, p=2e-105 between FACS and Kimmel, p=7e-69 between droplet and Kimmel).

2. Using GAGs identified in our FACS data, we compute the GAG score and further estimate the GAG score effects for all five data sets (TMS FACS data, TMS droplet, the bulk data from Schaum et al., Kimmel et al., Kowalczyk et al.). We found the chronological age effects on GAG score to be significantly positive in all five data sets, confirming that the GAG score captures the aging process.

3. The tissue-cell GAG score effects estimated from the five data sets are highly consistent.

4. Additionally, we showed that the GAG score is robust to different thresholds for selecting the global aging genes.

Please see the “Validating the GAG score on external data” section in the revised paper for more details.

4. For comparing aging effects purpose, I don't understand why aging scores of cells from the young time point should be included in the final aging score for each tissue-cell type. Firstly, it doesn't represent the aging effect if you also include scores from young cells; Secondly, the scores for each tissue-cell type may be confounded by different percentages of young cells.

Thank you for this question. To address the concerns raised by the reviewer, we performed the same analysis, namely the tissue-cell GAG effects estimation via regression, using only the old cells (18m/24m). We found that the result to be highly consistent. Please see Supp. Figure 13H. Here, each dot corresponds to a tissue-cell type, x-axis corresponds to the tissue-cell GAG score effects estimated using all cells, and y-axis corresponds to the GAG score effects estimated using only old cells.

Reviewer #2:In the manuscript titled "Mouse Aging Cell Atlas Analysis Reveals Global and Cell Type Specific Aging Signatures", Zhang and et al. systematically explored the aging-related genes in 76 tissue-cell types from 23 tissues in 10 male and 6 female mice from the Tabula Muris Senis single-cell transcriptomic dataset. The authors used a linear regression model to identify a set of up-/down-regulated aging-related genes, found a general down-regulation of gene expression in most tissue-cell types and revealed sets of aging regulated genes that shared by different tissue-cell types or that specifically enriched in some tissue-cell types. The authors further leveraged the average expression difference in the up- and down-regulated global aging genes, and proposed an aging score defined as a correlated factor that determines the change in different tissue-cell types without effects of age and sex. The manuscript is well organized, the results presented are impressing and the conclusions are pretty attractive. However, the data analyses need substantial revisions to draw any convincing conclusions, specifically, I have a list of concerns as below:1. General concerns:a. The sample size may be insufficient (10 males, 6 females) to build linear regression model to determine reliable correlations between certain gene expression with the two factors of age and sex in single-cell level.

Thank you for the question. Our study is actually one of the largest single-cell RNA-seq aging studies, containing 110k annotated cells from 16 mice in the FACS data and 250k annotated cells from 23 mice in the droplet data. Other aging studies have relatively fewer cells or fewer animals, e.g., 55k cells from 7 mice in [Kimmel et al. Genome Research 2019] and 50k cells from 16 mice in [Ximerakis et al. Nat. Neurosci. 2019]. Moreover, the statistical power of our analyses is substantially enhanced due to the fact that we have a large number of sequenced cells from diverse tissue-cell types in each animal. In our study, we considered each cell as a sample, which enabled us to have sufficient samples for fitting the linear regression model. Because most single-cell RNA-seq studies have a limited number of animals, it is a common and accepted practice to treat each cell as a sample [Finak Genome Biology 2015]. We have added a clarification on this in the revision.

b. Since the TMS datasets are curated from a collaboration program with different labs, confounding batch effects should be explicitly evaluated and resolved before any further analysis.

Thank you for the question. The TMS data was centrally collected and processed at the CZ Biohub, and does not contain significant batch effects as verified in the original TMS data generation paper. Specifically, the original paper showed that cells from the same cell type would cluster together by unbiased whole-transcriptome Louvain clustering (Extended Data Figure 1g-h [The Tabula Muris Consortium Nature 2020], https://www.nature.com/articles/s41586-020-2496-1/figures/5).

c. Though many platforms and technologies may have built-in strategies to deal with uncertainty of RNA copy number, normalization of single cell transcriptome are still highly recommended in the data processing to control technical inconsistency and calculation of DGE. The MAST package is aware of the problem and use the CDR calculation as alternatives of normalization, which, however, was omitted by the authors when using the package.

Thank you for the question. We agree with the reviewer that it is important to control for technical factors. We performed the size factor normalization the same as the original TMS paper (please see the “Data preprocessing” subsection in Methods).

We decided that it is better to not adjust for CDR because our analysis shows that CDR is positively correlated with age and negatively correlated with technical covariates such as sequencing depth and the number of detected ERCC spike-ins, both when considering all cells or considering each sex separately (Supp. Figure 1). As a result, controlling for CDR may remove genuine aging effects. This is why we presented the main results without CDR adjustment.

We also performed parallel analyses after adjusting for CDR and found that the age coefficients estimated with and without CDR correction were highly correlated (0.89 for the TMS FACS data and 0.93 for the TMS Droplet data, Supp. Figure 2). This demonstrates that our results are robust to CDR correction.

d. Beside the validation of the aging score, parallel analyses of droplets datasets should be performed in the same tissue-cell types to validate their findings.

Thank you for this helpful suggestion. We have added parallel experiments for the droplet data as validations, including DGE analysis (Supp. Figures 1,2,4), aging trajectory analysis (Supp. Figure 6), pathway enrichment analysis (Supp. Figures 10A-B), all GAG score (global aging gene score) analysis (the “Validating the GAG score on external data” section in the main paper), and tissue-level validations (Supp. Figures 17-19). We found the results are consistent and added comments in the corresponding places in the paper.

2. Identification of aging-related genes:a. In the linear model of DGE with age and sex, it is unclear what the observations were in the model, whether the statistical test performed in individual cells across different tissue-cell types; how to deal with missing data; and how to control the effect of cell numbers if the tests are performed in tissue-cell types level.

Thanks for the questions; we apologize that this was not clear in the original submission. We performed a DGE analysis for cells in each tissue-cell type separately. The samples correspond to all cells in the tissue-cell type and the observations correspond to the expressions of the gene across the cells. The zero counts in scRNA-seq were handled by the MAST package and we do not observe other types of missing data. Since the tests were performed at a cell level for each tissue-cell type, the cell numbers will only affect the detection power for each tissue-cell type.

Therefore, it is not a confounding factor and does not need to be controlled. We have clarified these in the revision.

b. FDR calculation is highly depend on simultaneous consideration of comparisons in multiple tests. It is also unclear what the exact number of comparisons considered in each FDR test.

The number of comparisons correspond to the number of genes in the tissue-cell types. We have clarified this in the revision.

c. Figure 1B shows number comparison between up- and down-regulated genes related to age regardless of sex, however the authors should clarify whether the sex has effect on the result and a new figure of the comparison considering sex is highly recommended.

Thanks for the good suggestion. We also performed the same DGE analysis using only cells from one sex or using cells from a subset of age groups (3m/18m). Please see Supp. Figures 3-4 for the new results.

Specifically, Figure 1B showed that most tissue-cell types have more down-regulated aging-related genes than up-regulated aging-related genes. We found that such a pattern was mostly driven by 24m mice and was not specific to one sex. We also observed a similar pattern on the droplet data.

3. Tissue-cell level global aging markers:a. the definition of global aging genes is based on an arbitrary cutoff of half of tissue-cell types without considering the data structure, tissue-cell similarities or proportions of cells determined in mice, which may be conceptually confusing if change happened as data accumulating and new tissue-cell types identified.

We have improved the analyses to better account for the data structures that you mentioned. We instead use the weighted tissue-cell types to determine the global aging genes, where the tissue-cell weights are inversely proportional to the number of cell types in the tissue to ensure equal representation of different tissues. Please see the “Global aging genes” subsection in Methods for more details.

We also performed a set of perturbation analysis to show that our results are not sensitive to variations of the global aging gene selection rules. Specifically, we have showed that:

1. Global aging genes selected using different significance thresholds give similar GO enrichment analysis results (Supp. Figure 10C).

2. Global aging genes selected using different significance thresholds and directionality thresholds give similar aging scores (i.e., the GAG scores, Supp. Figures 13A-D).

b. For those up- or down-regulated genes in specific tissue-cell types, a pickup top 20 genes is not convincing and it is necessary to evaluate the significance of the involvement of those genes in shared biological progresses or functions.

In the initial submission, we performed a canonical pathway enrichment analysis for global aging genes using the IPA software (previous Figure 2D, revision Supp. Figure 9B-C). In the revision, we also added a GO biological process pathway enrichment analysis using g:Profiler (Figure 2D) and performed KEGG pathway analysis using GSEA (GSEA MGSig Database, Supp. Figure 9A). The GO analysis showed that the global aging genes are significantly associated with apoptosis, translation, biosynthesis, metabolism, and cellular organization (each corresponding to multiple pathways with FDR<1e-4). The result is supported by the canonical pathway analysis and the KEGG pathway analysis. These biological processes are highly relevant to the aging process and are shared across most cell types, consistent with the intuition that GAGs represent the global aging process across tissue-cell types. In addition, the IPA canonical pathway analysis also highlighted the mTOR pathway, which is known to be highly relevant to aging. Based on these analyses, we have improved the discussion of the shared biological processes and function of aging related genes in the “Bimodal effects of aging and global aging genes” section of the main paper.

4. Aging score based on global aging genes:a. The aging score is defined based simply on the potential determinants in the average expression changes between up- and down-regulated global aging genes without the effects from age and sex, which is not intuitively straightforward and confusing, given the ambiguous involvement in biological functions or regulations of the list of genes.

Thank you for the question. As described in the response to the previous comment (3b), we have shown that the global aging genes are associated with aging-related biological processes through three sets of gene set enrichment analysis.

Moreover, to make the analysis more conceptually clear, we have defined a fixed effect model to jointly model the effects of the chronological age, sex, and tissue-cell types on the global aging gene (GAG) score:

GAG score ~ chronological age + sex + tissue-cell-type

Then we estimate the tissue-cell GAG score effects via regression (please see the GAG score subsection in Methods for more details). The tissue-cell GAG score effects capture how the GAG score of cells from the tissue-cell type deviates from the average GAG score of cells in the same age and sex group. Intuitively, a larger GAG score effect suggests that the corresponding cell type could be molecularly more sensitive to aging compared to other cells in the same animal.

We replicated all of our original findings using this estimated tissue-cell type GAG effects. First, we find that different tissue-cell types have heterogeneous effects on the GAG score, following the same ordering as in our original submission (Original Figure 3A, Revision Figure 3A). Second, stem cells and immune cells have higher effects while parenchymal cells have lower effects (Original Supp. Figure 8, Revision Figure 3B). Third, short-lived cells have higher GAG score effects (Original Figure 3B, Revision Figure 3C); and the estimates are consistent between different datasets (Original Figure 3C, Revision Figure 3D). These results demonstrate that we can reproduce and support our findings with this more principled statistical model of aging scores.

As validations, we added two additional data sets for comparison: Kimmel et al. GR 2019 on mouse aging in lung, kidney, and spleen, Kowalczyk et al. GR 2015 on mouse aging in subtypes of HSCs. We validate our GAG score analysis by showing:

1. The global aging genes identified in the Tabula Muris Senis (TMS) FACS data and the TMS droplet data significantly overlap with each other and both sets of genes significantly overlap with the 261 shared aging genes in Kimmel et al. (p=9e-48 between FACS and droplet, p=2e-105 between FACS and Kimmel, p=7e-69 between droplet and Kimmel).

2. Using GAGs identified in our FACS data, we compute the GAG score and further estimate the GAG score effects for all five data sets (TMS FACS data, TMS droplet, the bulk data from Schaum et al., Kimmel et al., Kowalczyk et al.). We found the chronological age effects on GAG score are significantly positive in all five data sets, confirming that the GAG score tags the aging process.

3. The tissue-cell GAG score effects estimated from the five data sets are highly consistent.

4. Additionally, we showed that the GAG score is robust to different thresholds for selecting the global aging genes.

Please see the “Validating the GAG score on external data” section in the revised paper for more details.

b. The authors claim that aging score is correlated with the cell turnover but failed to show any solid evidence.

Instead of correlating the tissue-cell GAG score effects (corresponding to the tissue-cell aging score residual in the previous submission) and the cell turnover rate as done in the initial submission, we assigned a binary cell lifespan label to each tissue-cell type and compared the tissue-cell GAG score effects between the long-lived cells and short-lived cells. This procedure allows us to incorporate more cell types where the exact lifespan information is not available but are commonly considered as long-/short- lived cells. We found short-lived cells have significantly higher GAG score effects as compared to long-lived cells (p=1.68e-3, Figure 3C)

5. Tissue-cell-specific aging genes:a. The authors grouped tissue-cell types into 6 categories based on annotated functionality, and then identified over-represented genes in each category as tissue-cell-specific aging genes. I am wondering why not define the tissue-cell-specific aging genes in original tissue-cell types; the arbitrary grouping may be inaccurate and inevitably reduced the resolution of the tissue-cell specificity.

Thanks for this helpful comment. As suggested by the reviewer, we further investigated aging-related genes that are specific to (1) a functional category, (2) a cell type, (3) a tissue, (4) a tissue-cell type. In order to do this, we designed a new method to identify category-specific genes based on meta-analysis (please see the “Category-specific aging genes” subsection in Methods for details).

1. For functional categories, genes specific to endothelial cells, immune cells, stem cells and stromal cells show strong specificity to the category they belong to and are enriched in highly relevant biological pathways. Genes specific to epithelial cells and parenchymal cells show a less specific pattern.

2. For cell types, we found genes specific to B cells, basal cells of epidermis, endothelial cells, macrophages, mesenchymal stem cells, mesenchymal stem cells of adipose, myeloid cells, and skeletal muscle satellite cells. We found that they are associated with relevant biological processes.

3. For tissues, we found genes specific to each tissue, but their functionalities are less interpretable.

4. We did not find any gene that has an aging signature specific to just a single tissue-cell type.

Please in the “Category-specific aging genes” section, Figure 4 and Supp. Figures 14-16 for more details.

b. Again, the author failed to establish solid connection between these tissue-cell-specific aging genes and the tissue-cell specific functions or aging progress, which makes the result less reliable.

Similar to the validation for global aging genes, we also performed GO biological pathway enrichment analysis for category specific genes. We found that functional-category specific genes and cell-type specific genes are associated with highly relevant biological processes. Please see Figure 4C and Supp. Figure 16B for more details.

[Editors’ note: what follows is the authors’ response to the second round of review.]

Essential revisions:1. GAG score aims to capture common aging states across all cell types. Though authors tried to validate it from different datasets, it would be more convincible if they could show how well GAG score is fitted for GAG score ~ chronological age + sex + tissue-cell-type with both scatter plot and proportion of explained. Alternatively, will it fit better for GAG score ~ chronological age + sex + tissue-cell-type + tissue-cell-type * chronological age, where tissue-cell-type could also affect the slope of age.

Thank you for this comment. As suggested by the reviewer, we added Figure 3—figure supplement 4C to report the proportion of variance explained by the original model and the alternative model with the interaction terms between age and each tissue-cell type. As shown in this new figure, the original model explains 60.2% of the GAG score variance in the TMS FACS data, indicating a reasonable fit. Adding the tissue-cell-type * chronological age interaction term in the model only slightly increased the model fit (explained variance 62.6%). In addition, we find the amount of improvement by the alternative model is similar across datasets. We added this result in the main paper.

It was difficult to visualize the multivariate model as scatter plots—for example the scatter plot of one of the covariates with the GAG score doesn’t capture the joint regression. Therefore, we did not include it.

2. GAG score, due to its nature of capturing global pattern, may lose the power to identify the aging state of those cell types with the specific program associated with aging. It may be the reason for observations mentioned in line 135-138 and 146-148, specifically in Sup.Figure 11, for Brain.Non-Myeloid. neuron, liver.hepatocyte, mamary_glad.bascal cells, cells of 24 months even have a smaller GAG score than cells of 18 month. However, This may be potentially explained by catogery-specific aging genes later identified. Would tissue-cell type with more category-specific aging genes be more likely to show this heterogenous aging state pattern? Further, an accurate aging score for a specific cell type may be better consisting of two parts, global aging score, and specific aging score.

Thank you for the comment. As you suggested, we investigated whether tissue-cell types with more category-specific aging genes are more likely to show heterogenous aging patterns, and the data does not show a consistent association. For example, parenchymal cells, including neurons and hepatocytes as mentioned in your comment, in general have a more heterogeneous aging process pattern. However, as shown in Figure 4B, there are relatively fewer genes specific to parenchymal cells as compared to other categories.

As a side point, the mamary_glad.basal cells as mentioned in your comment only have cells from two time points (3m and 18m), and do not exhibit a heterogeneous aging pattern, namely 24m cells having smaller GAG scores than 18m cells. We removed the 24m label in the corresponding figure legend (Figure 3—figure supplement 1B, mamary_glad.basal cell panel) to avoid confusion.

More generally, while the current data allows us to construct a global aging score (GAG score), it is more difficult to construct cell-type-specific aging scores, which requires cataloging cell types at a higher granularity. For example, an immune-specific aging score would require a comprehensive catalog of different types and states of immune cells in different tissues and organs. Overall, we identified much more global aging genes (330) compared to tissue-cell type specific aging genes (14 to 76). Comprehensive characterization of tissue-cell type specific aging scores may be out of the scope of our current data, which is why we focus on the global aging score. We agree with you that this is an exciting direction of future work and we added a discussion on this in the Discussion section.

3. Last session of tissue-level section is not interesting to me, since differential genes at cell type are already accurately defined, and there's no reason to go back to bulk differential signal, which may be confounded by other factors such as cell-type proportions.

Thanks for the comment. As suggested by the reviewer, we have moved the tissue-level validation section to the supplementary material and added the following remark in the main paper.